# Phytochemical Profile, Free Radical Scavenging and Anti-Inflammatory Properties of *Acalypha Indica* Root Extract: Evidence from In Vitro and In Vivo Studies

**DOI:** 10.3390/molecules26206251

**Published:** 2021-10-15

**Authors:** Ravi Sahukari, Jyothi Punabaka, Shanmugam Bhasha, Venkata Subbaiah Ganjikunta, Shanmugam Kondeti Ramudu, Sathyavelu Reddy Kesireddy, Weibing Ye, Mallikarjuna Korivi

**Affiliations:** 1Department of Zoology, Sri Venkateswara University, Tirupati 517502, India; sravijpl@gmail.com (R.S.); jaanusvu@gmail.com (J.P.); shanmugambmsc@gmail.com (S.B.); doctorsvenkat@gmail.com (V.S.G.); sathyakreddy56@gmail.com (S.R.K.); 2Department of Zoology, PRR &VS Government Degree College, Vidavalur 524318, India; krshanmugamphd@gmail.com; 3Exercise and Metabolism Research Center, College of Physical Education and Health Sciences, Zhejiang Normal University, Jinhua 321004, China

**Keywords:** C-reactive protein, edema, free radical scavengers, phytochemicals, Indian Acalypha

## Abstract

In our in vitro and in vivo studies, we used *Acalypha indica* root methanolic extract (AIRME), and investigated their free radical scavenging/antioxidant and anti-inflammatory properties. Primarily, phytochemical analysis showed rich content of phenols (70.92 mg of gallic acid/g) and flavonoids (16.01 mg of rutin/g) in AIRME. We then performed HR-LC-MS and GC-MS analyses, and identified 101 and 14 phytochemical compounds, respectively. Among them, ramipril glucuronide (1.563%), antimycin A (1.324%), swietenine (1.134%), quinone (1.152%), oxprenolol (1.118%), choline (0.847%), bumetanide (0.847%) and fenofibrate (0.711%) are the predominant phytomolecules. Evidence from in vitro studies revealed that AIRME scavenges DPPH and hydroxyl radicals in a concentration dependent manner (10–50 μg/mL). Similarly, hydrogen peroxide and lipid peroxidation were also remarkably inhibited by AIRME as concentration increases (20–100 μg/mL). In vitro antioxidant activity of AIRME was comparable to ascorbic acid treatment. For in vivo studies, carrageenan (1%, sub-plantar) was injected to rats to induce localized inflammation. Acute inflammation was represented by paw-edema, and significantly elevated (*p* < 0.05) WBC, platelets and C-reactive protein (CRP). However, AIRME pretreatment (150/300 mg/kg bodyweight) significantly (*p* < 0.05) decreased edema volume. This was accompanied by a significant (*p* < 0.05) reduction of WBC, platelets and CRP with both doses of AIRME. The decreased activities of superoxide dismutase, catalase, glutathione reductase and glutathione peroxidase in paw tissue were restored (*p* < 0.05 / *p* < 0.01) with AIRME in a dose-dependent manner. Furthermore, AIRME attenuated carrageenan-induced neutrophil infiltrations and vascular dilation in paw tissue. For the first time, our findings demonstrated the potent antioxidant and anti-inflammatory properties of AIRME, which could be considered to develop novel anti-inflammatory drugs.

## 1. Introduction

Inflammation is a complex biological immune response to harmful stimuli of irritant chemicals, pathogens and damaged local injury cells in the tissue. Inflammatory system plays an important role in life-threatening diseases including COVID-19, cancer, diabetes and cardiovascular dysfunction [1,2]. Leukocyte activation such as macrophages, neutrophils, eosinophils and monocytes may occur during inflammation in order to protect the tissue, and simultaneously trigger the production of reactive oxygen species (ROS) or free radicals [3]. These ROS, include superoxide anion (O_2_^−^), hydrogen peroxide (H_2_O_2_), and hydroxyl radical (^•^OH) are highly reactive in nature, which counteract with adjacent polyunsaturated fatty acids, proteins and DNA, and leading to cause lipid peroxidation, protein degradation, and DNA damage, respectively [4,5]. In order to combat this oxidative stress, cells are equipped with endogenous antioxidant enzymes and non-enzymatic defense systems. The antioxidant enzymes, mainly superoxide dismutase (SOD), catalase (CAT), glutathione reductase (GR) and glutathione peroxidase (GPx) however reach insufficient during inflammation and/or oxidative stress [4,6]. C-reactive protein is an acute phase inflammatory marker that remarkably elevates in various inflammatory diseases, including COVID-19, cancer, cardiovascular diseases (CVDs) and diabetes [7,8]. Oxidative stress can trigger various genes involved in inflammatory pathways, including nuclear factor kappa B (NF-κB), and intrinsically involved in the progression of chronic diseases [2,4]. It is therefore necessary to supplement anti-inflammatory and/or antioxidant compounds to assist the impaired immune and antioxidant defense systems.

Various plant-derived or synthetic antioxidant and anti-inflammatory compounds have been used to treat oxidative insult and inflammation, as these substances are capable in scavenging of ROS and inflammatory mediators [9,10]. However, the synthetic compounds have been claimed to cause mild to severe adverse effects. On the other hand, usage of herbal medication has been increased over the past three decades, and ~80% of the world’s population living in the developing nations are rely on plant-based medicinal products as the primary source of healthcare [11]. Although several herbal compounds have been widely used as antioxidant or anti-inflammatory substances, many of them are yet to be investigated and their molecular mechanism needs to be disclosed. Therefore, identification of novel and effective therapeutic substances from medicinal plants with low or no side effects became prioritized research. The phenolic compounds and flavonoids are a family of bioactive constituents have been demonstrated as promising source for new medications to treat inflammation and related chronic painful diseases [12]. Phytol, a diterpene constituent of chlorophyll has been shown to suppress pro-inflammatory cytokines release through downregulation of NF-κB signalling pathway [13]. Methanolic extracts *Sideritis bilgeriana* reported to suppress tumour necrosis factor alpha (TNF-α), interleukin-1β (IL-1β) and leukocyte migration in carrageenan-induced pleurisy model. These pharmacological effects are due to the presence of phenolic and flavonoids content detected by HPLC-DAD-UV analysis [14].

*Acalypha indica* also known as ‘Indian copperleaf’ or ‘Indian mercury’ is a medicinal plant that belongs to *Euphorbiaceae* family. *A. indica* is widely distributed throughout the Asia and Africa. In folk medicine, the whole plant and leaves of *A. indica* have been used to treat asthma, pneumonia, stomach discomforts, skin injuries and snake bites [15,16]. Leaves of *A. indica* have been scientifically tested for their anti-bacterial [17], anti-inflammatory [18] and antioxidant [19] properties. Few studies isolated and identified the phytochemical constituents from the leaf extracts of *A. indica* [16,19]. Stems of *A. indica* are also reported to have antioxidant, glucose lowering and hepato-protective properties in experimental diabetic rats [20].

Nevertheless, the pharmacological properties of *A. indica* root methanolic extract (AIRME) have not been demonstrated. Furthermore, isolation and characterization of phytochemical compounds in AIRME is yet to be performed. We assume that roots of *A. indica* may also contain various phytochemical constituents that could exhibits pharmacological efficacies. Therefore, we aimed to perform HR-LC-MS and GC-MS techniques to identify the bioactive compounds in AIRME. In vitro studies were performed to investigate the free radical scavenging activity of AIRME. Subsequently, in vivo studies were conducted on rats to evaluate the antioxidant and anti-inflammatory activities of AIRME in carrageenan-induced acute inflammatory model.

## 2. Results

### 2.1. Phytochemicals in A. indica Root Extract

Qualitative estimation of phytochemicals based on color intensity in the AIRME revealed that it has several phytochemical compounds such as phenols, flavonoids, saponins, tannins, reducing sugars, alkaloids, terpenoids, coumarins, anthraquinones and anthocyanins. We then quantified major phytochemicals, phenols and flavonoids based on qualitative analysis, which aided in further fractionation, isolation and biological assessment. Quantitative estimations revealed total phenolic content in AIRME was 70.92 mg of gallic acid equivalent/g extract, and flavonoids content was 16.01 mg of rutin/g extract. 

### 2.2. Phytochemical Profile of A. indica Root Methanolic Extract

After quantification of phytochemicals, the AIRME was subjected to GC-MS and HR-LC-MS for profiling their individual chemical constituents. GC-MS analysis revealed that the extract contains a significant amount of lipid soluble constituents (volatile) and their metabolites. To be specific, GC-MS results showed 14 volatile compounds whereas HR-LC-MS explored 101 semi volatile and non-volatile phytochemicals in AIRME. The details of each phytochemical constituent, including molecular formula, mass, PubChem CID and percentage were presented in Table 1. The predominant phytomolecules in AIRME through HR-LC-MS were, retusoquinone (1.694%), ramipril glucuronide (1.563%), antimycin A (1.324%), propionylglycine methyl ester (1.18%), swietenine (1.134%), quinone (1.152%), oxprenolol (1.118%), choline (0.847%), bumetanide (0.847%) and fenofibrate (0.711%). Similarly, the major phytochemical constituents by GC-MS were piperidine-2,5-dione (3.73%), cephalotaxine, 3-deoxy-3,11-epoxy-,(3a,11a)- (2.66%) and octadecanoic acid (2.053%). Most of these compounds displayed a higher number of hydrogen bond donor/acceptor counts. The chromatograms obtained from GC-MS and HR-LC-MS analyses (Appendix A) and chemical structures of all compounds identified in AIRME (Appendix A) were provided as Appendix A.

Computational calculations for the adverse effect descriptors of AIRME compounds showed that many compounds may not possess mutagenicity, tumorigenicity, reproductive toxicity and irritation (Table 1). These predictive non-toxic effects of the compounds need to be further confirmed through laboratory studies. 

### 2.3. A. indica Root Extract Scavenges DPPH and Hydroxyl Radicals

In vitro free radical scavenging activity of AIRME was determined by measuring the inhibition of DPPH and ^•^OH. The results revealed that AIRME (10–50 µg/mL) significantly scavenges the DPPH radical in a concentration dependent manner with a IC_50_ of 15.85 µg/mL, whereas the IC_50_ with ascorbic acid treatment was 5.71 µg/mL (Table 2). Hydroxyl radical-induced deoxyribose sugar damage was significantly decreased by AIRME in a concentration-dependent manner with IC_50_ of 14.74 µg/mL (Table 2). These findings signify the potent free radical scavenging activity of AIRME. 

### 2.4. A. indica Root Extract Suppress Hydrogen Peroxide and Lipid Peroxidation In Vitro

We found that the AIRME (20–100 µg/mL) effectively converted the hydrogen peroxide into water with IC_50_ value of 65.40 µg/mL whereas ascorbic acid converted at IC_50_ of 33.31 µg/mL. Furthermore, anti-lipid peroxidation activity of AIRME (20–100 µg/mL) was evidenced by a substantial suppression of lipid peroxides in egg homogenate. Inhibition of lipid peroxidation was proportionate with AIRME concentrations, and IC_50_ was 79.70 µg/mL. The efficacy of AIRME against hydrogen peroxide and lipid peroxidation propagation was comparable to the potency of ascorbic acid (Table 2).

### 2.5. A. indica Root Extract Decreases Paw Edema Volume in Rats

Carrageenan induced acute inflammation was evidenced by the appearance of paw edema in rats. Carrageenan-induced rat paw edema is a well-established animal model to evaluate the anti-inflammatory properly of phytochemicals [10,21]. The edema volume was sharply increased 1 h after injection (38%) and reached maximum size by 5/6 h (52%) (Figure 1). However, edema volume was significantly decreased with ARIME (150 and 300 mg/kg b.w.) pretreatment. Although both doses of AIRME significantly inhibited the edema growth against carrageenan, the high-dose of AIRME (300 mg/kg b.w.) is more effective in inhibiting (*p* < 0.05) the edema development. The edema volume with AIRME high-dose at 1, 2, 3, 4, 5 and 6 h after injection was 24, 32, 34, 29, 21 and 7%, respectively, which is significantly lower compared to carrageenan group at all time points (Figure 1). Our results further emphasized that the decreased edema volume with AIRME was parallel with the standard anti-inflammatory drug, diclofenac treatment. Physical observations of paw further revealed that carrageenan-induced inflammatory edema was smaller in AIRME pretreated rats than that of their disease controls (Figure 2).

### 2.6. A. indica Root Extract Attenuates Acute Inflammatory Response in Rats

Hematological analyses showed a significant rise in white blood cells and platelets in carrageenan injected group compared with normal control (*p* < 0.05), which indicates acute inflammation in rats. The surge in WBC count and platelets in inflammatory rats was not seen in AIRME pretreated rats. This was represented by a remarkably less number of WBC and platelets with low-dose (150 mg/kg; *p* < 0.05) and high-dose (300 mg/kg; *p* < 0.01) of AIRME treatment (Table 3). Similarly, elevated CRP level (55.94 ± 4.86 ng/mL) with carrageenan injection was substantially decreased in AIRME treated groups (low-dose 36.98 ± 3.24; high-dose 31.84 ± 3.09 ng/mL). The dose-dependent decrease of CRP with AIRME was statistically significant (*p* < 0.05 and *p* < 0.01) compared with disease control group (Table 3).

### 2.7. A. indica Root Extract Restores Antioxidant Enzyme Activities in Paw Tissue

We then studied the response of antioxidant system with AIRME treatment against acute inflammation. Carrageenan-induced impaired antioxidant homeostasis in paw tissue was represented by a significant reduction (*p* < 0.05) of SOD, CAT, GR and GPx enzyme activities. It is worth to note that carrageenan-induced loss of antioxidant enzyme activities was not seen in AIRME pretreated groups (150 and 300 mg/kg body weight). The dose-dependent preservation of antioxidant enzyme activities with AIRME was significant compared to the disease control group. To be specific, antioxidant property of the high-dose of AIRME is more effective (*p* < 0.01) and almost similar to the standard anti-inflammatory drug, diclofenac (Figure 3).

### 2.8. A. indica Root Extract Prevent Tissue Architectural Damage

Tissue architectural derangements that occurred upon carrageenan injection, and tissue protective effects with AIRME pretreatment against inflammatory damage were examined through histopathological studies. Images from normal control and AIRME control groups showed normal architecture of the paw tissue with normal muscle around decalcified bone (DB) without vascular dilation and neutrophil infiltration. Carrageenan-induced acute inflammation was visualized by redness and swelling of paw, and tissue derangements were represented by edema, vascular dilation and accumulation of neutrophil infiltration in disease control group (Figure 4). Nevertheless, AIRME pretreatment (150 and 300 mg/kg b.w.) confined the paw tissue architectural damage through preventing the neutrophil infiltration, vascular dilation and suppressing edema (Figure 4). The anti-edematous activity of AIRME against acute localized inflammation was comparable with diclofenac treatment.

## 3. Discussion

Previous studies have shown the antioxidant, anti-inflammatory and tissue protective properties of *A. indica* leaf and stem extracts [18,19,20]*,* but not the root extract. Besides, no study has identified the phytochemical constituents in *A. indica* root methanolic extract (AIRME) yet. For the first time, we screened the phytochemical constituents in AIRME, and reported 101 and 14 compounds from HR-LC-MS and GC-MS techniques, respectively. The major phytomolecules in AIRME were ramipril glucuronide (1.563%), antimycin A (1.324%), swietenine (1.134%), quinone (1.152%), oxprenolol (1.118%) choline (0.847%), bumetanide (0.847%) and fenofibrate (0.711%). Then, we demonstrated the antioxidant and anti-inflammatory properties of AIRME through in vitro and in vivo studies. In vitro free radical scavenging activity of AIRME was evidenced by a potent inhibition of DPPH and ^•^OH radicals and substantial suppression of H_2_O_2_ and lipid peroxidation. Evidence from in vivo studies revealed that carrageenan-induced paw edema was significantly decreased (volume) in AIRME pretreated rats. The surge in WBC and platelets with response to acute inflammation was significantly suppressed in AIRME pretreated groups. Furthermore, elevated CRP levels with carrageenan injection were not seen in AIRME pretreated rats, which indicates potent anti-inflammatory activity of AIRME. Most importantly carrageenan-induced impaired antioxidant homeostasis in paw tissue was attenuated with AIRME treatment. This was represented by a significant conservation of SOD, CAT, GPx and GR activities against carrageenan-induced loss. Histopathological studies further emphasize the tissue protective effect of AIRME against carrageen-induced tissue derangements, accumulation of neutrophil infiltration and vascular dilation.

Oxidative stress is an imbalance between pro-oxidants and antioxidants that is typically involved in the progression of many diseases, including cancer, CVDs, diabetes and atherosclerosis [22]. Natural antioxidant compounds, such as plant-based phenols and flavonoids can give protection against oxidative stress by scavenging the highly reactive free radicals, and inhibiting the lipid peroxidation [23]. In this study, naturally occurred large quantities of phenols (70.92 mg of gallic acid/g) and flavonoids (16.01 mg of rutin/g) in AIRME could potentially scavenge the free radicals, and thereby prevent the oxidative stress. HR-LC-MS is a chromatography technique that examines the polar (phenols and flavonoids) and semi-polar compounds in plant extract [19]. Furthermore, GC-MS analysis reported 14 compounds wherein non-polar/volatile phytochemicals are found in the root extract. Hydroxyl groups play an important role as a donor of hydrogen bonds, which is necessary for free radical scavenging, reduction of metals and molecular interaction with biomolecules [24]. In this study, we counted the hydrogen bond donors for all the identified phytochemicals through HR-LC-MS and GC-MS. The results showed that the majority of the compounds identified by HR-LC-MS exhibited these groups; while the number of compounds identified by GC-MS was lower (Table 1). These findings revealed that AIRME is a good source of antioxidant compounds, which could effectively eradicate the free radicals and/or prevent the oxidative stress.

Free radicals, and free radical source H_2_O_2_, is a well-known weak oxidizing agent, rapidly cross the cell membrane, reacts with ions such as Fe^2+^/Cu^2+^ and resulting in the generation of highly toxic hydroxyl radicals [25]. The cellular antioxidant defense system plays a significant role in quenching the toxic free radicals, and thereby protects the biomolecules from oxidative modification. In addition to the cellular antioxidant mechanism, the exogenous antioxidants from natural medicinal plants can enhance the protection against disease or stress [26,27]. The free radical scavenging activity of AIRME in this study was revealed by efficient scavenging of H_2_O_2_, ^•^OH and DPPH radicals, which is comparable to the scavenging ability of the standard ascorbic acid (Table 2). The existence of phenols and flavonoids with HBDC, such as Bumetanide, Bussein’s hydrolysis product, Coproporphyrin II, Dihydrodeoxystreptomycin, Ramipril glucuronide and Trandolapril glucuronide further strengthen the free radical scavenging or antioxidant activity of AIRME.

Identification and developing of natural medicinal compounds without adverse effects is critical for pharmaceutical companies in order to promote the medicinal values and to gain final marketing approval. Toxicity evaluation of the plant compounds is typically a laborious process with several complexities, and includes the requirement of animal models, cell lines, expenditure, man power and a long duration of time. To overcome these difficulties, computational calculation is an alternative approach to assess the adverse effects of newly identified plant compounds [28]. Therefore, we used computational calculation, and measured the potential adverse effects, such as mutagenicity, tumorigenicity, irritant and reproductive toxicity of the compounds that were identified in AIRME. The predictions showed that most of the compounds in AIRME may not have mutagenicity, tumorigenicity, irritant and reproductive toxicity. These toxicity assessments are predictive, and laboratory studies are necessary to confirm the toxicity of individual phytomolecules of AIRME.

Acute inflammatory response is mainly represented by an increased vascular permeability and neutrophil infiltration, which leads to edema formation. In the cellular events, extravasation of fluids and proteins is accompanied with excessive accumulation of leukocytes at the inflammatory site [29,30]. Carrageenan, a highly sulfated polysaccharide is widely used as food additive due to its capability to enhance the food structure [31]. Carrageenan (non-antigenic phlogistic agent)-induced rat paw edema is a well-established and widely used model to evaluate the anti-inflammatory properly of plant and synthetic compounds [10,21]. Carrageenan-induced edema development is a biphasic acute inflammatory response [29]. In the first phase from 0 to 2.5 h, carrageenan triggers the release of acute phase mediators, mainly serotonin, histamine and kinins, which influence the vascular permeability [32]. Then, prostaglandins are the key players in the second phase of inflammation that occurs around 3 h after carrageenan injection [33]. These events provoke the release of several pro-inflammatory mediators, where inhibition terminates the inflammatory process [34]. Local inflammation is associated with oxidative burst and increased production of pro-inflammatory cytokines, such as TNF-α, IL-1β, IL-6 and nitric oxide [35].

The carrageenan-induced inflammation in our study was witnessed by increased CRP levels and elevated WBC and platelet count. Platelets usually gathered at the injured site, adhere to the WBCs and release cytokines and chemokines that are chemotactic for neutrophils and monocytes. These events subsequently increase inflammation [36]. CRP is an acute phase inflammatory marker that increases rapidly and dramatically in response to acute inflammation [37]. It has been shown that CRP can enhance the platelet-activating factor (PAF)-induced inflammation via binding to PAF and its precursor [38]. One of our key findings is that AIRME received rats exhibited lower levels of CRP as well as decreased WBC and platelet count, which implies potent anti-inflammatory activity. In the phytochemical screening, we reported several pharmacological compounds, like oxtriphylline (0.677%), choline (0.847%), bumetanide (0.847%), swietenine (1.13%) and fenofibrate (0.711%) in the root extract. A previous study has shown that choline and aspirin together can suppress the pro-inflammatory mediators against acute inflammation induced by carrageenan and lipopolysaccharide [39]. Other phytochemicals, bumetanide [40], fenofibrate [41] and swietenine [42] are also reported to possess anti-inflammatory activity via downregulation of pro-inflammatory cytokines. Another study documented that methanolic extract *Sideritis bilgeriana* suppressed TNF-α, IL-1β and leukocyte migration in carrageenan-induced pleurisy model, which is due to the presence of f phenolic and flavonoids content [14]. It is persuaded that existence of these pharmacological compounds in AIRME could act as anti-inflammatory agents, and thereby suppress inflammation and edema in carrageenan-injected rats. The anti-inflammatory property of AIRME was comparable to the diclofenac, a nonselective non-steroidal anti-inflammatory drug.

ROS or free radicals at low concentrations act as signaling molecules; however, they are deleterious to cells at high concentrations. Due to their highly reactive nature, ROS reacts with vital biomolecules, proteins, lipids and DNA and can cause irreversible damage. To prevent such oxidative damage, cells have evolved an array of defense systems, including antioxidant enzymes that scavenge excessive ROS. Cells experienced oxidative stress when the balance tipped between antioxidants and free radicals [4,5]. All antioxidant enzymes, including SOD, CAT, GPx and GR estimated in paw edema were significantly decreased after carrageenan injection, which reflects impaired antioxidant homeostasis and state of oxidative stress. Carrageenan injection has been reported to trigger intracellular ROS production, decrease antioxidant capacity and cause inflammation [31,43]. Excessive production of oxygen free radicals in the extracellular space and relatively low activities of SOD and CAT in the inflammatory state increase the vulnerability of extracellular components to ROS and stimulate chemotaxis for other inflammatory cells [44].

However, pretreatment of AIRME restored the SOD, CAT, GPx and GR activities against carrageenan-induced loss. In vivo antioxidant property of AIRME is corroborates with in vitro DPPH radical scavenging activity, and suppressed H_2_O_2_ and lipid peroxidation. Similar to our findings, oral administration of amiodarone, an anti-arrhythmic agent to carrageenan-injected rats has been shown to decrease the paw edema volume and inflammation in association with increased antioxidant enzyme activities [45]. The anti-inflammatory property of *Pistacia lentiscus* fruit oil is accompanied by the increased activities of SOD, CAT and GPx and decreased lipid peroxidation in rat paw tissue after carrageenan injection [46]. Another study showed that cashew nuts supplementation ameliorated the carrageenan-induced diminution of SOD and CAT activities and GSH concentrations. Cashew nuts further suppressed the release of pro-inflammatory cytokines and prevent the oxidative stress [47]. These findings imply that the anti-inflammatory property of natural compounds is associated with antioxidant property against carrageenan-induced oxidative stress. The increased antioxidant enzyme activity or free radical scavenging capacity of AIRME in our study might be due to the presence of various phytochemicals in the root extract. AIRME phytochemical screening represented with high HBDC and HBAC, including ramipril glucuronide (1.563%), antimycin A (1.324%), dihydrodeoxystreptomycin (1.021%), swietenine (1.134%), hydrolysis product of bussein (1.145%), terbinafine metabolite (1.098), oxprenolol (1.118%) and 3-*O*-methylrimiterol (1.118%).

The antioxidant activity of AIRME was further confirmed by tissue protective effect against carrageenan-induced architectural damage to paw tissue. From the histopathological studies, it is visible that the inflamed area appears with edema, congestion, vascular dilation and neutrophil infiltration. These morphological changes in paw tissue are the typical events of local inflammation caused by carrageenan injection. However, AIRME treatment prior to carrageenan injection decreased edema and prevented vascular dilation and a surge in neutrophil infiltration, which reveals an anti-inflammatory property. Suppressed paw edema and neutrophil recruitment in paw tissue have been reported as a key anti-inflammatory property of crocin (an active constituent of saffron) against histamine injection in rats [48]. Naik and colleagues reported that pathological changes, including hepatic steatosis with severe swelling of hepatocytes and fat accumulation in obese rats were partially attenuated with *A. indica* leaf extracts. However, this study not revealed the phytochemical composition in the leaf extracts of *A. indica* [49]. Another study showed that quinone alkaloid pretreatment significantly inhibited carrageenan-induced paw edema in rats [50]. Astragalin, a naturally existing flavonoid in plants alleviated epithelial hyperplasia, severe leukocyte infiltration and sub epidermal edema induced by carrageenan injection [51]. AIRME also possesses flavonoids, quinine, phenols and other active ingredients which may contribute for the tissue protective effects. To our knowledge, this is the first evidence to demonstrate the tissue protective effect of AIRME against carrageenan-induced tissue damage.

## 4. Materials and Methods

### 4.1. Chemicals

The chemicals used in this research were obtained from the following scientific companies: Sigma (St. Louis, MO, USA), TCI (Shanghai, China), Merck (Mumbai, India), SRL (New Delhi, India) and Fisher (Pittsburgh, PA, USA).

### 4.2. Collection and Preparation of A. indica Root Extract

*A. indica* whole plant was collected from the campus garden of Sri Venkateswara University, Tirupati, India. Plant specimen was authenticated by the Taxonomist, Department of Botany, Sri Venkateswara University, Tirupati and herbarium was deposited with a voucher no: SVU-BOT-926. The roots were separated from the plant, and washed with running tap water to remove soil, then washed again with fresh water and shade dried. The dried root material was powdered using blender, and 100 g of the powder was transferred into amber color bottle containing 500 mL of methanol (1:5). The mixture was incubated with occasional stirring for 3 days, and the solvent was filtered with non-absorbent cotton, muslin cloth and Whatman no 1 filter paper finally. The filtrate was concentrated in rotary evaporator (BUCHI 036576, Roskilde, Denmark) and lyophilized (VIRTIS:Model BT4KZL-105, Midland, Canada) to obtain *A. indica* root methanolic extract (AIRME).

### 4.3. Qualitative Estimation of Phytochemicals

We have primarily tested for the phytochemicals of phenols, flavonoids, saponins, tannins, reducing sugars, alkaloids, terpenoids, coumarins anthraquinones and anthocyanins in the AIRME according to the protocols stated by Shukla and colleagues [52].

### 4.4. Quantitative Estimation of Phytochemicals

#### 4.4.1. Determination of Total Phenolic Content

According to Gutierrez and Navarro [53], the total phenolic content of AIRME was measured using the Folin Ciocalteu (FC) method. Briefly, 0.5 mL (5 mg) of extract at different concentrations (5–50 µg/mL) and same concentrations of standard gallic acid was added to 1 mL of FC (10%) reagent. For this reaction, 2.5 mL of sodium carbonate solution was added and incubated at room temperature for 30 min. Using a UV-Vis spectrophotometer (Shimadzu UV-1800, Kyoto, Japan) the resulting reaction was measured at 760 nm. The gallic acid standard graph was used to calculate the phenolic content of the extract, and expressed as mg of gallic acid equivalent per g of extract.

#### 4.4.2. Determination of Total Flavonoids Content

Total flavonoid content in AIRME was quantified by following aluminum chloride method [54]. Briefly, 1.0 mL of extract (10 mg/mL) was added to 4 mL of distilled water. To this, 0.3 mL of 5% NaNO_2_ and 0.3 mL of 10% aluminum chloride (AlCl_3_) solutions were added and incubate for 6 min. Then, 2 mL of 1 M NaOH solution was added and made up to 10 mL with distilled water. The reaction mixture was incubated for another 15 min and the flavonoid content was measured at 510 nm. Rutin was used as standard for calculating flavonoid content in AIRME, and content was expressed as mg of rutin equivalent per g of extract.

#### 4.4.3. Phytochemical Profiling and Gas Chromatography-Mass Spectrometry (GC-MS) analysis of AIRME

GC-MS analysis for AIRME was achieved by JEOL GC MATE ΙΙ (GC model, Agilent Technologies 6890N Network GC system, Santa Clara, CA, USA) equipped with HP 5 MS column. High pure helium as carrier gas at a constant flow rate of 1 mL/min was used for GC separation. Injector temperature was set at 220 °C and oven temperature was set as 50 °C raised to 250 °C at 10 °C/min. Total GC running time was 30 min. A highly sensitive quadruple double focusing mass analyzer was used and equipped with photon multiplier tube as the detector; mass range of 50 to 600 amu; and ionization voltage (Electron impact ionization) 70 eV was used. This protocol was also explained in our previous study [19]. Mass of each peak was obtained from mass spectroscopy while the proposed structures of compounds were predicted from the screening library of National Institute of Standard and Technology (NIST, Gaithersburg, MD, USA).

#### 4.4.4. Phytochemicals Profiling and HR-LC-MS Analysis of AIRME

The AIRME was then analyzed in HR-LC-MS (Model: Agilent 1290 Infinity UHPLC System, Santa Clara, CA, USA) in order to identify the finger prints of other hidden phyto moieties. For this assessment we followed procedure described by Ravi et al., [19]. Briefly, an auto sampler injected a 3 µL volume of the sample into a C18 column (ZORBAX 2.1 9. 50 mm 1.8 Micron). The auto sampler has a 100 µL/mL auxiliary pull and ejects the sample quickly. Two binary pumps (G4220B) are integrated in one housin and deliver desired mobile phase gradient ratios (solvent A: 0.1% Formic Acid (FA) in water and B: 90% Acetonitrile + 10% H_2_O + 0.1% FA). The total run time was 30 min, during which the solvent A-95% + B-5% was initiated at 2 min, the solvent A gradient ratio changed to 5% at 20 min, and the solvent A 95% was maintained from 26 to 30 min. The binary pump pressure was kept stable at 1200 bar, and the flow rate was 0.300 mL/min. The Mass Hunter workstation program was used to gain an edge in identifying correct MS and MS/MS for the extract’s LC profile.

#### 4.4.5. Computational Calculations for the Identified Phytochemicals

The resulted phytochemicals from analytical techniques were structurally drawn and their chemical descriptors, such as hydrogen bond donor count (HBDC) and hydrogen bond acceptor count (HBAC) were predicted using Marvin Sketch 16.3.21 version (https://chemaxon.com/products/marvin, (accessed on 8 June 2021)), a chemo informatics tool. In addition, the pharmacological properties of these compounds include tumorigenic, mutagenic, irritant and reproductive effects were assessed using DataWarrior V4.7.2 software (http://www.openmolecules.org/datawarrior/, (accessed on 8 June 2021)).

### 4.5. In Vitro Studies

Free Radical Scavenging and Anti-Lipid Peroxidation Activities of AIRME

In vitro free radical scavenging activity of AIRME was carried out by incubating different concentrations of extract (10–50 µg/mL) with 0.6 mM DPPH. The resulting reaction optical density (OD) was read at 517 nm against ethanol blank [55]. Hydroxyl radical (^•^OH) scavenging ability of AIRME was assayed by adopting the deoxyribose method as described by Nagai et al., [56]. The ^•^OH radicals of Fentons reagent degrade the deoxyribose sugar, whereas prevention of such degradation of sugar by AIRME was studied with different concentrations (10–50 µg/mL). The OD of samples was read at 520 nm against blank containing total reaction mixture with absence of AIRME.

The hydrogen peroxide (H_2_O_2_) inhibition/reduction property of AIRME was carried out by incubating different concentration of extract (20–100 µg/mL) with H_2_O_2_ for 30 min. Retained H_2_O_2_ after incubation was read at 230 nm against phosphate buffer blank [57] in UV-VIS spectrophotometer (Shimadzu UV-1800, Kyoto, Japan). Next, anti-lipid peroxidation property of AIRME was determined using egg homogenate treated with Fentons reagent for 1 h incubation as described by Upadhyay et al. [58]. Prevention of lipid peroxidation by various concentrations of AIRME (20–100 µg/mL) was read at 532 nm using total reaction without extract as a blank.

### 4.6. In Vivo Studies

#### 4.6.1. Experimental Design and Treatment

Anti-inflammatory and antioxidant activity of AIRME was studied in 1% carrageenan induced paw edema in rats. Thirty-six male adult rats of Wistar strain were purchased from the Indian Institute of Science (IISc), Bangalore. Rats were kept in cages, allowed to acclimatize for a week with conditions of 25–28 °C temperature, 50–55% humidity, with regular light/dark cycle (7:00 AM to 7:00 PM). Strict ethical guidelines for the maintenance and care of experimental animals were followed. The entire study design and protocols used in this study were reviewed and approved by the Institutional Animal Ethics Committee (09/(i)/a/CPCSEA/IAEC/SVU/ZOOL/KSR/Dt.08/07/2012) of Sri Venkateswara University, Tirupati, India. After acclimatization, rats (190 gm ± 10 gm bodyweight) were divided into six experimental groups with six rats in each as mentioned below:Group I—Normal control (NC): Rats in this group received 50 µL normal saline by sub-plantar injection (SPI);Group II—Disease control (DC): Carrageenan (1%, 100 µL) was injected (SPI) into the rat paw to induce acute localized inflammation;Group III—AIRME control (AIRME): Rats in this group received only AIRME (300 mg/kg bodyweight) by oral gavage;Group IV—AIRME low-dose plus carrageenan (AIRME-L+C): Rats were orally administered with low dose of AIRME (150 mg/kg body weight) through oral gavage 1 h prior to carrageenan injection.;Group V—AIRME high-dose plus carrageenan (AIRME-H+C): Rats in this group received high dose of AIRME (300 mg/kg body weight) 1 h prior to carrageenan injection, as described in group IV;Group VI—Standard treatment (ST): Rats in this group received diclofenac (20 mg/kg body weight) by oral gavage one hour before carrageenan injection.

Carrageenan-induced rat paw edema is a well-established and widely used model to evaluate the anti-inflammatory properly of plant and synthetic compounds [10,21,59]. The AIRME doses (150 and 300 mg/kg body weight) used in this study were based on our preliminary dose-dependent studies, where we found various doses of AIRME (100, 150, 200, 250 and 300 mg/kg bodyweight) did not produce any physiological or behavioral changes in normal rats. Besides Sathya and colleagues documented no adverse with different doses of *A. indica* ethanolic extracts ranges from 5 to 2000 mg/kg b.w. [60].

#### 4.6.2. Measurement of Paw Volume

The perimeter of the inflamed and AIRME treated paws of experimental animals was measured using digital screw gauge (model: Insize-3109-25s-digital outside micrometer) as described by Killari and team (2019) [61]. The paw volume was measured for every hour until 6 h following carrageenan injection. The percent increase in paw thickness was calculated using the formula (Equation (1)):(1)$$\frac{Y_{t}-Y_{0}}{Y_{0}x\times 100}$$
where:$Y_{t}$—thickness of paw at different time intervals (1, 2, 3, 4, 5 and 6 h) (after carrageenan induction);$Y_{0}$—thickness of paw at 0 h (before injection of carrageenan).

Resulted final value for every group at every hour was obtained through calculated mean.

#### 4.6.3. Blood Collection and Analyses

After treatment period, rats were subjected to chloroform anesthesia and sacrificed with cervical dislocation. On the experiment day, the blood samples (approximately 5 mL) were immediately collected through heart puncture into tubes that contain anti-coagulant (EDTA). In blood, white blood cells (WBC) and platelets were counted using a haemogram. For the CRP assay, blood samples were centrifuged at 4000× *g* rpm for 15 min, and serum was collected. Then C-reactive protein (CRP) levels were measured in the serum using kit provided by Aspen Laboratories (Himachal Pradesh, India). The changes in CRP in different groups were expressed as ng/mL. The blood collection procedures and assay protocols were approved Institutional Animal Ethics Committee.

#### 4.6.4. Determination of Antioxidant Enzyme Activities in Paw Tissue

Paw tissue was separated, cleaned with normal saline and stored at −40 °C for further biochemical analysis. Antioxidant status of paw tissue was measured by assessing the primary antioxidant enzyme activities. The activity of superoxide dismutase (SOD) in the paw homogenates was measured using the Misra and Fridovich [62] method at 480 nm for 4 min on a Shimadzu UV-1800 spectrophotometer. The activity was measured as the quantity of enzyme that prevents epinephrine from being oxidized by 50%, which was equal to 1 U per mg of protein. Catalase (CAT) activity was evaluated using Aebi, [63] procedure, and the sample’s absorbance was recorded using a UV spectrophotometer at 240 nm for 1 min. The moles of hydrogen peroxide (0.066 M) decomposed per mg of protein each minute equals to one unit activity. The GR activity was measured at 340 nm for 3 min in spectrophotometer. The GR activity was expressed in micromoles of NADPH oxidized per mg of protein per minute [64]. GPx activity of paw tissue was measured using a reaction mixture composed of GSH (0.01 M), 12 mM *t*-butyl-hyroperoxide, 1.5 mM NADPH, and GR (0.24 units). The mM of NADPH oxidized/mg protein/min was measured at 340 nm for 180 s to determine enzyme activity [65].

#### 4.6.5. Histopathology

The paw tissue biopsy of rats from all experimental groups were taken and fixed in 10% formalin to prevent autolysis, which permanently cross-links and stabilizes their proteins. Then, the paw tissue was decalcified with weak formic acid before trimming to acquire the proper size and orientation of tissue (10%). The tissue sample was also infiltrated with paraffin to obtain its water content before being embedded in paraffin. Then the tissue embedded in paraffin was sliced into 6–8 μm section cuttings in a microtome as described by Slaoui and Fiette [66]. Further, tissue sections were stained with hematoxylin and eosin (H&E) on glass slide. The histopathological changes in paw tissue sections were observed under the microscope (Trinocular BX53, Olympus, Singapore) and the changes were recorded at 40× magnification.

### 4.7. Statistical Analysis

We used SPSS (Version 15, SPSS Inc., Chicago, IL, USA) and Excel for statistical analyses. The data expressed as means ± standard deviation (SD). One-way analysis of variance (ANOVA), followed by Duncan’s multiple comparison tests were performed for group comparison and significance. Statistical significance was set at *p* < 0.05.

## 5. Conclusions

For the first time, we demonstrated that *A. indica* root extract is rich in various phytochemical constituents without adverse effect descriptors. The major phytomolecules are ramipril glucuronide, antimycin A, swietenine, quinone, oxprenolol, choline, bumetanide and fenofibrate. AIRME showed potent free radical scavenging activity (DPPH and ^•^OH) and inhibition of lipid peroxidation ability. AIRME pretreatment diminished carrageenan-induced paw edema, and decreased WBC, platelets and CRP levels in rats. Improved antioxidant status with AIRME was further confirmed by its tissue protective effects against acute inflammatory damage. Owing to the potent antioxidant and anti-inflammatory properties of AIRME, our findings suggest considering of AIRME in developing and preparation of novel anti-inflammatory drugs. However, further studies are necessary to explore in-depth molecular mechanism with specific known compounds of *A. indica* root extract.

## Figures and Tables

**Figure 1 molecules-26-06251-f001:**
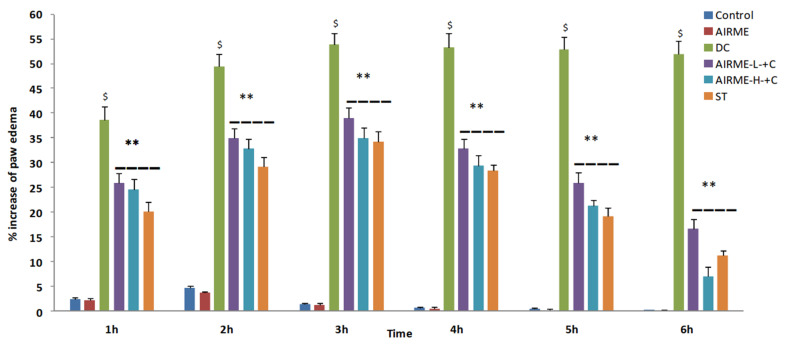
Percent changes in paw volume during the time course of 6 h after carrageenan injection (acute inflammation) in control and AIRME pretreated groups. Values are significant compared with control ($ *p* < 0.05) and compared with carrageenan-injected group (** *p* < 0.01). NC, normal control; DC, disease control; AIRME, *A. indica* root methanolic extract; AIRME-L+C, AIRME-low-dose plus carrageenan; AIRME-H+C, AIRME-high-dose plus carrageenan; ST, standard treatment.

**Figure 2 molecules-26-06251-f002:**
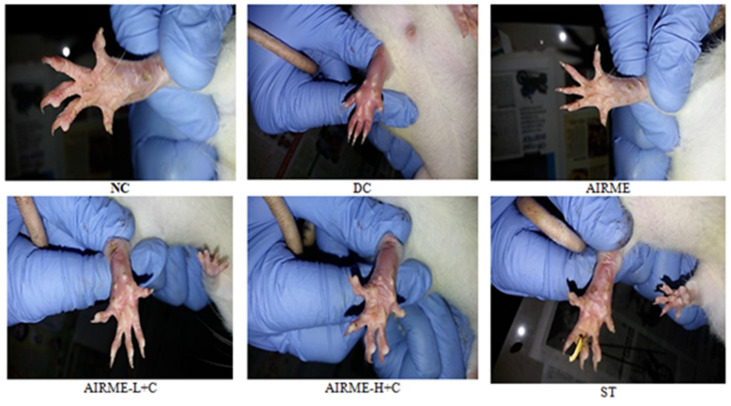
Physiological changes in rat paw tissue following carrageenan injection (1%) in different groups with or without AIRME 150. mg/kg), AIRME-H+C (300 mg/kg) and diclofenac (ST) pretreated rats. NC, normal control; DC, disease control; AIRME, *A. indica* root methanolic extract; AIRME-L+C, AIRME-low-dose plus carrageenan; AIRME-H+C, AIRME-high-dose plus carrageenan; ST, standard treatment.

**Figure 3 molecules-26-06251-f003:**
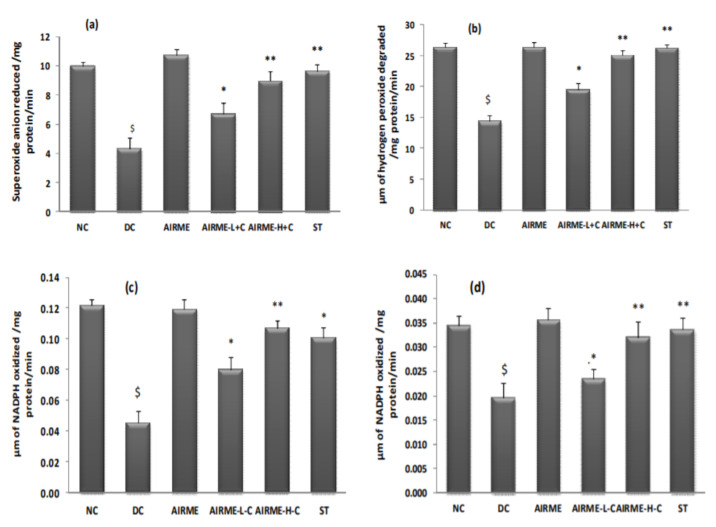
Changes in antioxidant enzyme (superoxide dismutase (**a**), catalase (**b**), glutathione reductase (**c**), glutathione peroxidase (**d**)) activities in paw tissues of different groups of rats. NC, normal control; DC, disease control (carrageenan); AIRME, *Acalypha indica* root methanolic extract; AIRME-L+C (AIRME low dose plus carrageenan); AIRME-H+C (AIRME high dose plus carrageenan), and ST (standard treatment, diclofenac). All values represented as mean ± SD (*n* = 6). $ indicates significant (*p* < 0.05) difference with NC. * (*p* < 0.05) and ** (*p* < 0.01) indicates significant difference with DC.

**Figure 4 molecules-26-06251-f004:**
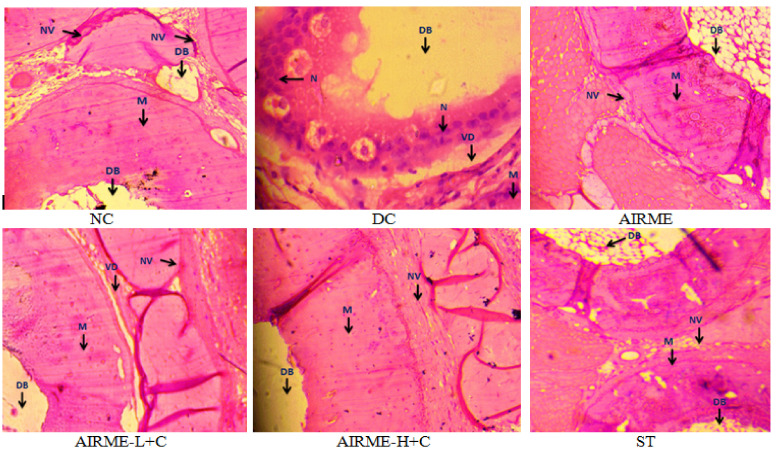
Histopathological study of inflamed paw tissue treated with AIRME. Paw tissue in NC and AIRME groups showed normal appearance of muscle (M) around the decalcified bone (DB), no vascular dilation (NV) and no infiltrate.

**Table 1 molecules-26-06251-t001:** Phytochemical profile of AIRME using chromatography techniques (HR-LC-MS and GC-MS), antioxidant descriptors (Hbds—Hydrogen bond donor count and Hbas—Hydrogen bond acceptors count) and adverse properties (Mut—mutagenicity, Tum—tumarogenicity, Irri—irritant and RepTox—reproductive toxicology) predicted using Marvin sketch, a computational tool.

No	Compounds Identified by HR-LC-MS	RT	Formula	Mass	Hbds	Hbas	Mut	Tum	Irri	RepTox	PubChem CID	Percentage
1	Oxtriphylline	0.42	C_12_H_22_N_5_O_3_	284.1715	1	4	No	No	No	No	656652	0.677
2	Choline	0.501	C_5_H_14_NO	104.1074	1	2	No	No	No	No	6209	0.847
3	Bumetanide	0.612	C_17_H_20_N_2_O_5_S	364.1112	3	7	No	No	Yes	No	2471	0.847
4	Indospicine	0.612	C_7_H_15_N_3_O_2_	173.1165	4	4	No	No	Yes	No	108010	0.776
5	10-Hydroxy-2*E*,8E-Decadiene-4,6-diynoic acid	0.613	C_10_H_12_O_3_	180.0772	2	2	No	No	Yes	No	9543616	0.745
6	2,6-Piperidinedicarboxylic acid	0.615	C_7_H_11_NO_4_	173.069	2	4	No	No	No	Yes	557515	0.881
7	Ala Ser His	0.616	C_12_H_19_N_5_O_5_	313.1392	6	7	No	No	No	No	-	0.644
8	Arg Pro	0.656	C_11_H_21_N_5_O_3_	271.1652	5	7	No	No	No	No	-	0.677
9	His Gly Val	0.658	C_13_H_21_N_5_O_4_	311.1577	5	6	No	No	No	No	-	0.627
10	Pro Ile	0.662	C_11_H_20_N_2_O_3_	228.1459	4	4	No	No	No	No	-	0.661
11	Fenofibrate	0.669	C_20_H_21_ClO_4_	360.116	0	3	No	No	No	No	3339	0.711
12	Mycophenolic Acid	0.671	C_17_H_20_O_6_	320.1239	2	5	No	No	No	No	446541	0.881
13	Mimosine	0.673	C_8_H_10_N_2_O_4_	198.0642	3	6	low	No	No	No	440473	0.911
14	Dimethylglycine	0.684	C_4_H_9_NO_2_	103.0634	1	3	No	No	No	No	673	0.640
15	3,7-Epoxycaryophyllan-6-one	0.688	C_15_H_24_O_2_	236.1759	0	2	No	No	No	No	4330530	0.745
16	l-2-Aminoadipic acid	0.778	C_6_H_11_NO_4_	161.0686	3	5	No	No	No	No	92136	0.474
17	*Trans*-4-Hydroxy-L-proline	0.869	C_5_H_9_NO_3_	131.058	3	4	No	No	No	No	5810	0.372
18	*N*2-Succinylglutamic acid	0.883	C_9_H_13_NO_7_	247.0694	4	7	No	No	No	No	25244383	0.406
19	Mucronulatol((+/−))	1.157	C_17_H_18_O_5_	302.111	2	5	No	No	No	Yes	442811	0.372
20	Propionylglycine methyl ester	1.698	C_6_H_11_NO_3_	145.0738	1	2	No	No	No	No	-	1.186
21	Ile Tyr	1.744	C_15_H_22_N_2_O_4_	294.1575	4	6	No	No	No	No	-	0.847
22	Phendimetrazine	1.855	C_12_H_17_NO	191.131	0	2	No	No	No	No	30487	1.016
23	4-(2-Hydroxy-3-isopropylaminopropyl)benzoic acid	2.163	C_13_H_19_NO_4_	253.1313	3	5	No	No	No	No	-	0.406
24	Mexiletine	2.219	C_11_H_17_NO	179.1309	0	1	No	No	Yes	No	4178	0.406
25	*N*-(3-Oxo-octanoyl)homoserine lactone	2.753	C_12_H_19_NO_4_	241.1335	1	3	No	No	Yes	No	127293	0.474
26	Retusoquinone	3.071	C_11_H_12_O	160.0872	0	1	No	No	No	No	6111564	1.694
27	Peucenin	3.385	C_15_H_16_O_4_	260.103	2	4	No	No	No	No	68477	0.949
28	Dehydrovariabilin	3.494	C_17_H_14_O_4_	282.0845	0	3	No	No	No	Yes	624785	1.050
29	Normeperidine	3.598	C_14_H_19_NO_2_	233.1414	1	2	No	No	No	No	32414	0.881
30	Thr Ala Ser	3.605	C_10_H_19_N_3_O_6_	277.1311	6	7	No	No	No	No	-	1.016
31	Diethyl Oxalpropionate	3.711	C_9_H_14_O_5_	202.0842	0	3	No	No	No	No	97750	0.881
32	3-*O*-Methylrimiterol	3.763	C_13_H_19_NO_3_	237.1366	3	4	No	No	No	No	-	1.118
33	Citrulline *n*-butyl ester	4.092	C_10_H_21_N_3_O_3_	231.1582	3	3	No	No	Yes	No	6426971	0.542
34	2(Benzylmethylamino)Ethanol *N*-Oxide	4.126	C_10_H_15_NO_2_	181.1102	1	2	No	No	No	No	264996	0.372
35	Chloramphenicol alcohol	4.177	C_11_H_14_N_2_O_6_	270.0845	4	6	Yes	No	No	No	10107147	0.406
36	12-Aminododecanoic acid	4.41	C_12_H_25_NO_2_	215.1891	2	3	No	No	No	No	69661	0.745
37	3-Oxa-9azoniatricyclo[3.3.1.02,4]non ane, 9-ethyl-9-methyl-7[(phenylacetyl)oxy]	4.522	C_18_H_24_NO_3_	302.1757	1	4	No	No	No	No	-	0.542
38	3,11-Dihydroxy myristoic acid	4.835	C_14_H_28_O_4_	260.2002	3	4	No	No	No	No	5282923	0.881
39	Avocadene acetate	4.835	C_19_H_36_O_4_	328.2622	2	3	No	No	Yes	No	3624980	0.881
40	Sulindac sulfide	4.881	C_20_H_17_FO_2_S	340.0902	1	2	No	No	No	No	5352624	1.013
41	6-Methoxyquinoline	4.928	C_10_H_9_NO	159.0686	0	2	Yes	No	No	No	14860	1.288
42	Edrophonium	4.928	C_10_H_16_NO	166.1233	1	1	No	No	No	No	3202	1.220
43	Quinine	4.939	C_20_H_24_N_2_O_2_	324.1837	1	4	No	No	No	No	3034034	1.152
44	3-[2(Dimethylamino)ethoxy]phen ylmethyl]-4-methyl-phenol	5.11	C_18_H_23_NO_2_	285.173	1	3	No	No	No	No	-	1.050
45	Metyrapol	5.301	C_14_H_16_N_2_O	228.1292	1	3	No	No	No	No	161210	0.949
46	Oxprenolol	5.322	C_15_H_23_NO_3_	265.1682	2	7	No	No	Yes	No	4631	1.118
47	Oxyphencyclimine	5.541	C_20_H_28_N_2_O_3_	344.2094	1	6	No	No	No	No	4642	0.986
48	Oxycodone	6.166	C_18_H_21_NO_4_	315.1486	1	9	No	No	No	No	5284603	0.406
49	Ergoline-8-methanol, 10-methoxy-1,6-dimethyl	6.279	C_18_H_24_N_2_O_2_	300.1858	1	3	No	No	No	Yes	-	0.847
50	*Evoxine*	6.332	C_18_H_21_NO_6_	347.136	2	10	No	No	No	No	73416	0.542
51	2-Hydroxytrimipramine	6.339	C_20_H_26_N_2_O	310.2038	1	3	No	No	No	No	160610	0.474
52	Penbutolol	6.575	C_18_H_29_NO_2_	291.2196	2	5	Yes	No	No	No	37464	1.152
53	Ala Lys Ile	6.581	C_15_H_30_N_4_O_4_	330.23	7	10	No	No	No	No	-	1.322
54	Oxymorphone	6.76	C_17_H_19_NO_4_	301.131	2	9	No	No	No	No	5284604	0.334
55	Diamorphine (heroin)	6.762	C_21_H_23_NO_5_	369.1566	0	7	No	No	No	Yes	5462328	0.406
56	Terbinafine metabolite	6.911	C_19_H_23_NO_3_	313.1672	3	5	No	No	No	No	-	1.098
57	Strychnine	6.912	C_21_H_22_N_2_O_2_	334.1672	0	5	No	No	No	No	441071	1.050
58	Hydrolysis product of bussein	7.021	C_32_H_40_O_14_	648.241	8	26	No	No	low	No	54699399	1.145
59	Ambelline	7.026	C_18_H_21_NO_5_	331.1412	0	9	Yes	No	No	No	25092366	1.321
60	2-Pyrrolidinone, 4-(2aminoethyl)-1-ethyl-3,3diphenyl-(AHR 5904)	7.039	C_20_H_24_N_2_O	308.1882	2	3	No	No	No	No	164917	1.023
61	Glu Val Asp	7.221	C_14_H_23_N_3_O_8_	361.1518	7	17	No	No	No	No	-	0.953
62	10-Nitro,9*Z*,12*Z*octadecadienoic acid	7.339	C_18_H_31_NO_4_	325.2246	1	8	No	No	No	No	5282259	0.994
63	5β-Chol-2-en-24oic Acid	7.397	C_24_H_38_O_2_	358.2954	1	4	No	No	No	No	-	0.912
64	5α-Cholan-24oic Acid	7.733	C_24_H_40_O_2_	360.311	1	4	No	No	No	No	5283802	0.832
65	Etioporphyrin III	7.793	C_32_H_38_N_4_	478.317	2	4	No	No	No	No	141310	0.814
66	Coproporphyrin II	8.066	C_36_H_38_N_4_O_8_	654.2548	6	10	No	No	No	No	165072	0.964
67	Arg Ala Arg	8.473	C_15_H_31_N_9_O_4_	401.2552	15	15	No	No	No	No	-	0.456
68	(5*Z*)-4,4-difluoro1alpha,25-dihydroxyvitamin D3/(5*Z*)-4,4-difluoro1alpha, 25dihydroxycholecalcifer	8.766	C_27_H4_2_F_2_O_3_	452.304	3	6	No	No	No	No	-	0.754
69	Arg Val Pro	8.887	C_16_H_30_N_6_O_4_	370.2321	8	12	No	No	No	No	-	1.101
70	Dihydrodeoxystreptomycin	9.017	C_21_H_41_N_7_O_11_	567.2862	16	29	No	No	No	No	11953824	1.021
71	Swietenine	9.019	C_32_H_40_O_9_	568.2704	1	10	No	No	No	No	76327465	1.134
72	C16 Sphinganine	9.686	C_16_H_35_NO_2_	273.266	3	3	No	No	No	No	5283572	0.786
73	(*Z*)-*N*-(2hydroxyethyl)hexadec-7-	11.085		297.2658	2	2	No	No	No	No	-	0.354
74	*N*-Hexadecyl-l-hydroxyproline	11.684	C_21_H_41_NO_3_	355.307	3	4	No	No	No	No	66690736	0.834
75	Trihexyphenidyl *N*-oxide	12.227	C_20_H_31_NO_2_	317.2342	1	2	No	No	No	No	129318457	0.965
76	Zearalenone	12.772	C_18_H_22_O_5_	318.1478	2	4	No	No	No	No	5281576	0.236
77	GPEtn(10:0/11:0)[U]	12.813	C_26_H_52_NO_8_P	537.3399	2	4	No	No	No	No	-	0.875
78	25-Hydroxycholesterol(d3)	12.829	C_27_H_43_D_3_O_2_	405.3607	2	2	No	No	low	No	-	0.823
79	Disopyramide	13.729	C_21_H_29_N_3_O	339.2395	1	3	No	No	No	No	3114	0.787
80	12-Hydroxy-10octadecynoic acid	13.763	C_18_H_32_O_3_	296.2341	2	3	No	No	No	No	-	0.867
81	Anandamide (18:3, *n*-6)	14.295	C_20_H_35_NO_2_	321.2652	2	2	No	No	No	No	5283445	0.564
82	9-Hydroxy-12-oxo10-octadecenoic acid	14.4	C_18_H_32_O_4_	312.2287	2	4	No	No	No	No	5282968	0.753
83	13-OxoODE	14.417	C_18_H_30_O_3_	294.2184	1	3	No	No	Yes	No	5283012	0.700
84	2*R*-hydroxy9*Z*,12*Z*,15*Z*-octadecatrienoic acid	14.648	C_18_H_30_O_3_	294.2188	2	3	No	No	No	No	16061058	1.032
85	Lactone of PGFMUM	14.85	C_16_H_24_O_5_	296.1612	1	4	No	No	No	No	-	0.854
86	*N* Acetylsphingosine	15.403	C_20_H_39_NO_3_	341.2915	3	3	No	No	No	No	5497136	1.132
87	Phytosphingosine	16.285	C_18_H_39_NO_3_	317.2918	4	4	No	No	No	No	122121	0.126
88	Dihydroceramide C2	16.767	C_20_H_41_NO_3_	343.3073	3	3	No	No	No	No	6610273	0.453
89	3-Deacetyl Khivorin	16.926	C_30_H_40_O_9_	544.2667	1	5	No	No	No	No	46878892	0.763
90	Harderoporphyrin	17.931	C_35_H_36_N_4_O_6_	608.2594	5	8	No	No	No	No	3081462	0.998
91	His Arg Val	18.001	C_17_H_30_N_8_O_4_	410.2411	8	9	No	No	No	No	-	0.324
92	Harderoporphyrinogen	18.177	C_35_H_42_N_4_O_6_	614.3115	4	9	No	No	Yes	No	193825	0.231
93	Ramipril glucuronide	18.404	C_29_H_40_N_2_O_11_	592.2643	5	10	No	No	No	No	-	1.563
94	Deoxykhivorin	18.791	C_32_H_42_O_9_	570.282	0	4	No	No	No	No	6708722	0.232
95	Antimycin A (A1 shown)	19.021	C_27_H_38_N_2_O_9_	534.2592	3	6	No	No	Yes	No	12550	1.324
96	*N*-(2hydroxyethyl)icosanamide	19.051	C_22_H_45_NO_2_	355.3432	2	2	No	No	No	No	3787294	1.021
97	9,14,19,19,19Pentadeuterio-1α,25dihydroxyprevitamin D3 / 9,14,19,19,19-pentadeuterio1alpha,25	19.24	C_27_H_39_D_5_O_3_	421.355	1	1	No	No	low	No	-	1.240
98	2Tricosanamidoethanesulfonic acid	19.241	C_25_H_51_NO_4_S	461.3489	2	4	No	No	No	No	-	1.200
99	Trandolapril glucuronide	20.16	C_30_H_42_N_2_O_11_	606.2793	5	10	No	No	No	No	92023960	0.932
100	2,4,6-Trimethyl-2,15tetracosadienoic acid	20.184	C_27_H_50_O_2_	406.3821	1	2	No	No	No	No	-	0.901
101	3β,6α,7αTrihydroxy-5β-cholan-24oic Acid	20.754	C_24_H_40_O_5_	408.2845	4	5	No	No	No	No	-	0.875
**Compounds Identified by GC-MS**												
1	Piperidine-2,5-dione	8.97	C_5_H_7_NO_2_	113.047	1	2	No	No	No	No	533930	3.733
2	Phenol, 2,4-bis(1,1-dimethylethyl)-	12.53	C_14_H_22_O	206.167	1	1	No	No	No	No	528937	1.333
3	Tridecanoic acid,12-methyl-, methyl ester	14.92	C_15_H_30_O_2_	242.224	0	1	No	No	No	No	21204	1.200
4	1-Oxacyclopentadecan-2-one,15,15-dimethyl	15.63	C_16_H_30_O_2_	254.224	0	1	No	No	No	No	-	0.800
5	11-Hexadecenoic acid, methyl ester	16.88	C_17_H_32_O_2_	268.24	0	1	No	No	No	No	5364696	1.040
6	Pentadecanoic acid, 14-methyl-, methyl ester	17.08	C_17_H_34_O_2_	270.255	0	1	No	No	No	No	21205	1.493
7	Octadecanoic acid	17.78	C_19_H_38_O_2_	298.287	0	1	No	No	No	No	5281	2.053
8	13-Hexyloxacyclotridec-10-en-2-one	18.52	C_18_H_30_O_2_	278.224	0	1	No	No	No	No	5369119	1.066
9	10-Octadecenoic acid,methyl ester	18.82	C_19_H_36_O_2_	296.271	0	1	No	No	No	No	5364425	1.954
10	Octadecanoic acid,methyl ester	19.05	C_19_H_38_O_2_	298.287	0	1	No	No	No	No	8201	0.956
11	Dasycarpidan-1-methanol,acetate (ester)	20.37	C_20_H_26_N_2_O_2_	326.44	1	2	No	No	No	No	550072	0.543
12	Aspidospermidine-3-carboxylic acid, 2,3-didehydro-,methyl ester,[5a,12a,19a]-	20.65	C_21_H_26_N_2_O_2_	338.199	1	3	No	No	No	No	-	2.123
13	Cephalotaxine, 3-deoxy-3,11-epoxy-, (3a,11a)-	22.96	C_18_H_19_NO_4_	313.131	0	5	No	No	No	No	-	2.666
14	4-*H*-1-Benzopyran-4-one, 7-(acetyloxy)-2-(4-(acetyloxy)phenyl)-5-methoxy-	25.6	C_19_H_14_O_7_	354.073	1	5	Yes	No	No	Yes	-	2.456

**Table 2 molecules-26-06251-t002:** In vitro free radical scavenging activity of AIRME. The results were compared with ascorbic acid and values represented as mean ± SD (*n* = 3).

**Conc. (µg/mL)**	**Scavenging of DPPH Radical (%)**		**Scavenging of Hydroxyl Radical (%)**		**Conc. (µg/mL)**	**Inhibition of Hydrogen Peroxide (%)**		**Inhibition of Lipid Peroxidation (%)**	
	AIRME	Ascorbic Acid	AIRME	Ascorbic Acid		AIRME	Ascorbic Acid	AIRME	Ascorbic Acid
10	41.02 ± 0.58	50.38 ± 0.40	46.17 ± 0.77	53.81 ± 0.85	20	22.53 ± 1.41	42.68 ± 0.76	20.68 ± 1.72	33.06 ± 2.85
20	56.92 ± 1.15	61.46 ± 0.66	53.82 ± 1.13	61.88 ± 1.19	40	41.73 ± 0.70	55.88 ± 1.13	31.03 ± 2.97	43.34 ± 2.65
30	62.82 ± 0.59	70.36 ± 0.99	60.74 ± 1.61	68.48 ± 0.86	60	51.00 ± 1.17	59.44 ± 0.83	40.80 ± 2.10	50.77 ± 1.98
40	66.02 ± 0.80	77.04 ± 1.21	64.69 ± 1.30	82.36 ± 1.04	80	58.31 ± 1.09	62.49 ± 0.80	52.58 ± 1,87	58.66 ± 2.10
50	68.46 ± 0.70	78.88 ± 1.65	71.48 ± 0.63	84.41 ± 1.61	100	63.20 ± 1.20	68.85 ± 0.68	57.75 ± 2.06	65.25 ± 3.06
IC_50_	15.85	5.71	14.74	5.28	IC_50_	65.40	33.31	79.70	59.45

**Table 3 molecules-26-06251-t003:** Effect of AIRME on white blood cells (WBC) and platelets count, and changes in serum C-reactive protein (CRP) in rats. Values represented as mean ± SD (*n* = 6). $ denotes significant at *p* < 0.05 compared with normal control (NC). Results are significant at *p* < 0.05 (*) and *p* < 0.01 (**) compared with disease control (DC).

Groups	WBC (10^9^/L)	Platelets (10^9^/L)	CRP (ng/mL)
NC	3.95 ± 2.06	320 ± 4.94	28.93 ± 2.95
DC	14.48 ± 4.76 ^$^	1025 ± 8.67 ^$^	55.94 ± 4.86 ^$^
AIRME	4.02 ± 2.14	343 ± 5.07	30.32 ± 3.06
AIRME-L+C	9.76 ± 2.75 *	765 ± 6.06 *	36.98 ± 3.24 *
AIRME-H+C	4.89 ± 2.43 **	366 ± 5.51 **	31.84 ± 3.095 **
ST	4.37 ± 2.14 **	337 ± 5.06 **	30.94 ± 3.065 **

NC—normal control; DC—disease control (carrageenan); AIRME—*Acalypha indica* roots methanolic extract; AIRME-L+C—AIRME-low dose plus carrageenan; AIRME-H+C—AIRME-high dose plus carrageenan and ST—standard treatment (diclofenac).

## Data Availability

The data presented in this study are available from the corresponding authors on request.

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
