# Peer review of "Phytochemical Profile, Free Radical Scavenging and Anti-Inflammatory Properties of Acalypha Indica Root Extract: Evidence from In Vitro and In Vivo Studies"

_molecules, 2021, doi:10.3390/molecules26206251_

Round 1
Reviewer 1 Report
See attached file

Author Response
Responses to Reviewers’ Comments: molecules-1393172
Reviewer 1
We express our sincere thanks to the Reviewer for the critical evaluation and positive comments on our manuscript. We agree with the comments, and we have revised the manuscript accordingly. English grammar mistakes were corrected carefully and the final version of the manuscript was read and corrected by the language expert in the field. All the comments (mentioned on review forum and also in the attached PDF file) were addressed and appropriate corrections to the manuscript were marked in red color. Here we are providing a detailed response to each comment.
Comment: Please insert a new paragraph showing the importance of extract and the chemical profile of polyphenols, flavonoids, and natural products as a therapeutic potential anti-inflammatory and influence and correlations of the anti-inflammatory effect with cytokines. It’s observated that poorly correlated chemical compositions and the mechanism of action were cited or explored in this work. Among so many articles, Perhaps these articles are useful to be cited.
Cavalcanti, Mariana RM, et al. "HPLC-DAD-UV analysis, anti-inflammatory and antineuropathic effects of methanolic extract of Sideritis bilgeriana (lamiaceae) by NF-κB, TNF-α, IL-1β and IL-6 involvement." Journal of Ethnopharmacology (2020): 113338.
Carvalho, Alexandra MS, et al. "Phytol, a Chlorophyll component, produces antihyperalgesic, anti-inflammatory, and antiarthritic effects: possible NFκB pathway involvement and reduced levels of the proinflammatory cytokines TNF-α and IL-6." Journal of natural products 83.4 (2020): 1107-1117.
Quintans-Júnior, Lucindo J., et al. "Dereplication and quantification of the ethanol extract of Miconia albicans (Melastomaceae) by HPLC-DAD-ESI-/MS/MS, and assessment of its anti-hyperalgesic and anti-inflammatory profiles in a mice arthritis-like model: Evidence for involvement of TNF-α, IL-1β and IL-6." Journal of ethnopharmacology 258 (2020): 112938.
Response: Authors are thankful to the Reviewer for this suggestion. As suggested, we have inserted a new paragraph in the Introduction section (Page 2). According to the scope of our manuscript, the recommended references were also cited in the new paragraph. The new paragraph explored the importance of phytochemicals in inflammation and oxidative stress conditions.
Comment: The chemical profile identifications are very confused. Please separate the compounds present in figure S2 by the method of identification, so I suggest the authors that show the compounds that were identified by GC-MS and HRLC-MS-MS and apply this information for discussions of results.
Response: We appreciate Reviewer for this useful comment. As suggested, we separated the compounds presented in Figure S2 based on the methods. In the revised version, Figure 3S showed the chemical compounds identified by HRLC-MS-MS and Figure 4S showed the chemical compounds identified by GC-MS analysis.
Comment: What is the criterion used to determine of screening of dose for in vivo assay? Based in the antioxidant results? The dose is based in LD50? Explain.
Response: Based on our previous reports (unpublished) and other studies, we have chosen the given dose of AIRME (150 and 300 mg/kg bodyweight) in the present study, and determined the anti-inflammatory and antioxidant property.
For instance, a study conducted on normal rats showed no toxic effects of various doses of A. indica ethanolic extracts either in acute or sub-acute toxicity studies. In acute toxicity studies, rats treated with various doses of A. indica ethanolic extracts ranges from 5 mg/kg b.w. to 2000 mg/kg b.w. did not produce any toxicity, behavioral changes and mortality. In sub-acute toxicity studies, rats treated with different doses of A. indica ethanolic extracts that is 100, 200, 300, 400 and 500 mg /kg b.w. for 30 days, not produced any adverse effects on bodyweight, organ weights and hematological parameters (Hb, RBC, WBC, ESR). Besides, blood biomarkers of liver and kidney function, glucose, uric acid and calcium levels were not considerably altered by different doses of A. indica ethanolic extracts (Sathya et al., 2012). However, diseased rats showed positive response to various doses of A. indica. Naik et al., (Naik et al., 2019) used 200 mg/kg b.w. of methanolic extract of A. indica and demonstrated the anti-adipogenic activity in obese rats. Another study used 300-600 mg/kg b.w. of A. indica stem methanolic extract, and reported antioxidant, glucose lowering and hepato-protective properties in diabetic rats (Priya and Rao, 2016). In our preliminary study, unpublished data showed that different doses of A. indica root methanolic extract (100, 150, 200, 250 and 300 mg/kg) did not produce any adverse effects in normal rats. We observed normal behavior, without irritation, frequent urination, weakness, slow locomotion, and hair erection. Based on these reports, we selected 150 and 300 mg/kg b.w. to explore the anti-inflammatory and antioxidant properties of AIRME.
Now we have included appropriate information about the dosage in the revised manuscript, Page 13 and 14.
Comment: I suggest using the Pearson analysis to verify possible correlations between the total phenolic/flavonoid and antioxidant assays.
Response: Authors are thankful to the Reviewer for this comment. To our understanding, Pearson correlation requires a group of samples. However, here in our study, we analyzed the phytochemicals in only one fraction that is methanolic. Therefore, we are unable to show the correlations between phenols/flavonoids and antioxidant assays.
Comment: I suggest promoting the correlations between inflammations and chemical and biological antioxidant effects.
Response: We agree with the Reviewer’s opinion. As suggested, the correlations between inflammatory mediators and chemical/biological antioxidants were highlighted in the revised manuscript, Page 1 & 2 (first and second paragraphs) and also in Page 10 & 11.
Comment: Further, it is not clear how this study is relevant to using this plant. Then, the authors must highlight the real contribution of this paper concerning that previously published. What is the contribution to the state of the art? I suggest using a chemical profile to present and discuss the results from the perspective of future use.
Response: Authors would like to make it clear in two points. First one, carrageenan-induced acute inflammation in paw edema is a well established and widely accepted model to evaluate the anti-inflammatory property of plant or synthetic compounds. The second one, there are reports to demonstrate the antioxidant and anti-inflammatory properties of A. indica leaf and stem extracts (Priya and Rao, 2016;Ravi et al., 2017;Kameswari et al., 2020). However, there are no studies to demonstrate the anti-inflammatory and antioxidant properties of A. indica root methanolic extract. In relevance, we designed this study to investigate the anti-inflammatory and antioxidant property of AIRME in an acute inflammatory model of carrageenan-induced paw edema.
Then, we have explained the real contribution of this study concerning the previous reports in the first paragraph of the Discussion, Page 8. Similarly, the contribution to the state of art was also highlighted in the Introduction and Discussion sections wherever applicable.
As suggested, the major phytochemicals in AIRME was discussed with respect to their antioxidant and anti-inflammatory properties in the revised manuscript. Please note that the explanations were there in the Results (Page 3) and Discussions parts (Page 10 & 11) in accordance with the scope of the study. In addition to these, the suitable sentences about chemical profile were also incorporated in the Abstract and Conclusions (Page 15).
Comment: The quality of English writing throughout the manuscript needs minor adjusting. Native or professional English writer assistance may be required
Response: We are thankful to the Reviewer for this helpful comment. As recommended, the revised manuscript has been carefully checked and polished by the professional English writer.
Bibliography
Kameswari, S., Narayanan, A.L., and Rajeshkumar, S. (2020). Free radical scavenging and anti-inflammatory potential of Acalypha indica mediated selenium nanoparticles. Drug Invention Today 13.
Naik, R., Nemani, H., Pothani, S., Pothana, S., Satyavani, M., Qadri, S.S.Y.H., Srinivas, M., and Parim, B. (2019). Obesity-alleviating capabilities of Acalypha indica, Pergulari ademia and Tinospora cardifolia leaves methanolic extracts in WNIN/GR-Ob rats. Journal of Nutrition & Intermediary Metabolism 16, 100090.
Priya, C.L., and Rao, K.B. (2016). Postprandial antihyperglycemic and antioxidant activities of Acalypha indica Linn stem extract: an in-vivo study. Pharmacognosy magazine 12, S475.
Ravi, S., Shanmugam, B., Subbaiah, G.V., Prasad, S.H., and Reddy, K.S. (2017). Identification of food preservative, stress relief compounds by GC–MS and HR-LC/Q-TOF/MS; evaluation of antioxidant activity of Acalypha indica leaves methanolic extract (in vitro) and polyphenolic fraction (in vivo). Journal of Food Science and Technology 54, 1585-1596.
Sathya, M., Kokilavani, R., and Ananta Teepa, K. (2012). Acute and subacute toxicity studies of ethanolic extract of Acalypha indica Linn in male Wistar albino rats. Asian J Pharm Clin Res 5, 97-100.
Reviewer 2 Report
The topic of the manuscript is of high scientific importance. In general, the manuscript is well written. The research is scientifically grounded and performed systematically. However, several issues need to be resolved to improve the quality of the work presented.
The Introduction section should be rewritten to highlight the aim of the present study that is missing. The capture of Figure 1 should be presented with more clarity. Please explain the abbreviations (for example, AIRME-L+C (AIRME low dose plus carrageenan); AIRME-H+C (AIRME high dose plus carrageenan), and ST (standard treatment, diclofenac)). Also, some minor typographical and grammatical errors throughout the manuscript should be checked and corrected.
Author Response
Responses to Reviewers’ Comments: molecules-1393172
Reviewer 2
We express our sincere thanks to the Reviewer for giving the positive comments on our manuscript. We agree with the comments, and we have revised the manuscript accordingly. English grammar mistakes were corrected carefully and the final version of the manuscript was read and corrected by the language expert in the field. All the comments were addressed and appropriate corrections to the manuscript were marked in red color. Here we are providing a detailed response to each comment.
Comment: The Introduction section should be rewritten to highlight the aim of the present study that is missing.
Response: The entire Introduction part has been revised to highlight the aim and importance of the study. Additional one new paragraph was added to emphasize the importance of phytomolecules in maintaining the antioxidant and inflammatory systems.
Comment: The capture of Figure 1 should be presented with more clarity.
Response: We are thankful to the Reviewer for this important comment. As suggested the Figure 1 Line diagram was replaced with a bar diagram, which clearly explained the paw volume changes by time.
Comment: Please explain the abbreviations (for example, AIRME-L+C (AIRME low dose plus carrageenan); AIRME-H+C (AIRME high dose plus carrageenan), and ST (standard treatment, diclofenac)).
Response: Apologizing for not explaining the full forms of the abbreviations. This is actually happened as we moved the Methods section after Discussion (according to the journal’s format). Anyhow, all abbreviation are now fully explained.
Comment: Also, some minor typographical and grammatical errors throughout the manuscript should be checked and corrected.
Response: Authors appreciate Reviewer for this comment. All typographical and grammatical errors have been checked and corrected in the whole manuscript. Further, the revised manuscript has been proof read and corrected by the professional English expert.
Reviewer 3 Report
The article ''Phytochemical Profile, Free Radical Scavenging and An-2 ti-inflammatory Properties of Acalypha Indica Root Extract: 3 Evidence from In Vitro and In Vivo Studies'' is very interesting in the field of phytochemistry and pharmacology. But it has not been presented properly. Details are attached in the pdf file
The major pitfalls of this manuscript are:
The GC MS experiment did not carry out with the proper methods. The column condition and output results are totally abnormal since the authors run methanol extract.
The toxicological study presented by computation has no justification for toxicity study without laboratory data.
The in vivo methods were presented without any guidelines.
The mechanism of action presented very poorly.
The conclusion has not reflected according to the findings
Please take care of the article very carefully otherwise it may liable for rejection.

Author Response
Responses to Reviewers’ Comments: molecules-1393172
Reviewer 3
We express our sincere thanks to the Reviewer for his/her critical evaluation and positive comments on our manuscript. We agree with the comments, and we have revised the manuscript accordingly. English grammar mistakes were corrected carefully and the final version of the manuscript was read and corrected by the language expert in the field. All the comments (mentioned on review forum and also in the attached PDF file) were addressed and appropriate corrections to the manuscript were marked in red color. Here we are providing a detailed response to each comment.
Comment 1: The predominant phytochemicals content, and significant results of in vivo and vitro must include in the abstract.
Response: Authors are thankful to the Reviewer for this meaningful comment. As suggested we are now included the predominant phytochemicals in AIRME, and significant results of in vitro and in vivo were mentioned in the Abstract.
Comment 2: Phytochemicals in A. indica Root Extracts 82: Phenols and flavonoids results should express as in terms of Standard equivalent on dry basis
Response: We are thankful to the Reviewer for this suggestion. Now the results of phenols and flavonoids were expressed with their respective standard equivalents.
The phenolic content in AIRME expressed in terms of gallic acid equivalent that is 70.92 mg of GA/g dry weight, whereas flavonoids concentrations were expressed in terms of rutin equivalents that is 16.01 mg of RU/g dry weight of extract. Standard equivalent information has included in the revised manuscript wherever applicable, and marked in red color.
Comment 3: Phytochemical Profile of A. indica Root Methanolic Extract: Must mention the most predominant compounds
Response: As suggested, the predominant compounds identified in AIRME through HR-LC-MS and GC-MS were presented in the Results subsection 2.2., Page 3.
Comment 4: 2.2. Phytochemical Profile of A. indica Root Methanolic Extract: Computational predictions mutagenicity, tumorigenicity, reproductive toxicity and irritation do not support without carrying out toxicity studies. It depends on many factors. You can just predict but not confirm. Please clarify and amend the matter.
Response: We agree with the Reviewer’s opinion that we cannot confirm the toxicity of compounds through computational calculations. Instead, computational calculations can be applied to predict the toxicity of newly identified compounds using a chemo informatics tool and DataWarrior software. Although computational calculations can predict the non-toxicity of the identified compounds, laboratory experiments are necessary to confirm such non-toxic effects. Accordingly, we have amended our statements in the revised manuscript, Page 3 and 9.
Typically, toxicity evaluation of the plant compounds is a laborious process with several complexities, include requirement of animal models, cell lines, expenditure, man power and long duration of time. To overcome these difficulties, computational calculation is an alternative approach to assess or predict the adverse effects of newly identified plant compounds (Krewski et al., 2010). In silico toxicology is one type of toxicity assessment that uses computational methods to analyze, simulate, visualize, or predict the toxicity of chemicals. In silico toxicology aims to complement existing toxicity tests to predict toxicity, prioritize chemicals, guide toxicity tests, and minimize late‐stage failures in drugs design (Raies and Bajic, 2016). Therefore, we predicted toxicological characteristics for the identified phytomolecules.
Comment 5: 2.5. A. indica Root Extracts Decreases Paw Edema Volume in Rats 125: The graph is not clear. also the results presented very vaguely. You can read following articles to get more knowledge for improving your presentation. https://doi.org/10.1016/j.jep.2019.01.025: https://doi.org/10.1016/j.jep.2021.114182
Response: Authors are thankful to the Reviewer for this valuable comment and suggesting the useful references. Now we have revised the Figure 1, and presented as bar diagram which explain time-dependent changes in paw volume. The presentation of results also modified accordingly in the revised manuscript, Page 5.
Comment 6: Discussion: Mechanism of action did not discuss clearly. It should be more specific with strong logical arguments.
Response: As suggested the Discussion section has been improved. The revised Discussion is more specific and logical according to our findings reported in this study. Additional explanation of molecular mechanism with key signaling molecules involved in inflammation and antioxidant systems may better explain the whole story, but it will be out of the scope of our study. Therefore, we clearly discussed our findings based on the data reported in this study.
Comment 7: 4.4.1. Determination of Total Phenol Content: Phenols and flavonoids results should express as in terms of Standard equivalent on dry basis
Response: We are apologizing for not mentioning the standard units. As suggested, we have included the standard equivalents to both phenolic content and flavonoids per gram weight of extract. This information is there in subsection 4.4.1 and also in other appropriate places.
Comment 8: 4.4.1. Determination of Total Phenol Content: Phenols and flavonoids results should express as in terms of Standard equivalent on dry basis
Response: We are apologizing for not mentioning the standard units. As suggested, we have included the standard equivalents to both phenolic content and flavonoids per gram weight of extract. This information is there in subsection 4.4.2 and also in other appropriate places.
Comment 9: How you retrieve this method ? There is no proper description of the methods. The authors did not mention any reference guidelines.
Response: Authors would like to bring to Reviewer’s notice that carrageenan induced acute inflammatory model is well established and widely used to assess the anti-inflammatory properties of phytochemicals. Reviewer suggested references (DOIs above) also used the carrageenan induced inflammatory model.
We have already mentioned this information in the discussion part, and now provided suitable information about inflammatory model and dosage of AIRME with supporting references in the revised manuscript, Page 13 and 14.
Comment 10: 4.6.2. Measurement of Paw Volume: How you find this calculation? There is reference found
Response: We measured the paw volume as described by Killari and colleagues (2019) (Killari et al., 2019). Now the supporting reference has been included and calculation formula was clearly presented in the revised manuscript, Page 14.
Comment 11: Blood collection and Analysis: No guideline mentioned
Response: Authors are thankful to the Reviewer for letting us to add essential information. The required information about blood collection, and units of CRP levels were provided in the revised manuscript, Page 14.
Comment 12: 4.6.5. Histopathology 496: What method you follow?
Response: For the histopathology studies, we have followed the method described by Slaoui and Fiette (Slaoui and Fiette, 2011). Now the protocol was further explained and the reference was included in the revised manuscript, Page 14/15.
Comment 13: The conclusion presented very poorly. There is no perfect remarks along with advantage of this study.
Response: As suggested, the conclusions were revised with perfect remarks and application of the study was also explained.
Comment 14: GC Ms data are very doubtful. it is completely a wrong experiment. Because they analysed methanol extract. they did not mention any column condition and which column used. Most of the compounds are very poor amount which indicates the wrong methodology. Please repeat the experiment and present the result with RI Index by using homogeneous alkane series solvent eg C7-C40
Response: There is no need to be doubtful on HR-LC-MS and GC-MS analyses. We are doing these assays for the past few years. We further assure that these assays were performed at Sophisticated Analytical Instrument Facility (SAIF), Indian Institute of Technology (IIT), Madras (Chennai) and IIT Bombay (Mumbai), India. This information is now added under acknowledgments section. We agree with the Reviewer that some compounds are poor in amount, but that does not imply wrong methodology. For the Reviewer’s information here we are providing the detailed protocol, and appropriate corrections were also done in the revised manuscript, Page 12.
GC-MS analysis for AIRME was done by JEOL GC MATE ΙΙ (GC model, Agilent Technologies 6890N Network GC system, USA) equipped with HP 5 MS column. High pure helium as carrier gas at a constant flow rate of 1 ml/min was used for GC separation. Injector temperature was set at 220 °C and oven temperature was set as 50°C raised to 250 °C at 10 °C/min. Total GC running time was 30 min. High sensitive quadruple double focusing mass analyzer was used and equipped with photon multiplier tube as the de-tector; mass range of 50 to 600 amu; and ionization voltage (Electron impact ionization) 70 eV was used. This protocol was also explained in our previous study (Ravi et al., 2017). Mass of each peak was obtained from mass spectroscopy while the proposed structures of compounds were predicted from the screening library of National Institute of Standard and Technology (NIST, Maryland, USA).
Bibliography:
Killari, K., Prasad, K., Talluri, M., Bokam, Y., Nadiminti, S., and Kommavari, C. (2019). Antiinflammatory Activity of Wheat Grass Fortified with Cow Urine Distillate. Indian journal of pharmaceutical sciences 81, 521-526.
Krewski, D., Acosta, D., Andersen, M., Anderson, H., Bailar, J.C., Boekelheide, K., Brent, R., Charnley, G., Cheung, V.G., Green, S., Kelsey, K.T., Kerkvliet, N.I., Li, A.A., Mccray, L., Meyer, O., Patterson, R.D., Pennie, W., Scala, R.A., Solomon, G.M., Stephens, M., Yager, J., Zeise, L., Staff of Committee on Toxicity, T., and Assessment of Environmental, A. (2010). Toxicity Testing in the 21st Century: A Vision and a Strategy. Journal of Toxicology and Environmental Health, Part B 13, 51-138.
Raies, A.B., and Bajic, V.B. (2016). In silico toxicology: computational methods for the prediction of chemical toxicity. Wiley Interdisciplinary Reviews: Computational Molecular Science 6, 147-172.
Slaoui, M., and Fiette, L. (2011). "Histopathology procedures: from tissue sampling to histopathological evaluation," in Drug safety evaluation. Springer), 69-82.
Round 2
Reviewer 1 Report
No more comments.
Reviewer 3 Report
Accept in present form